# Neural Machine Translation with Universal Visual Representation

**Zhuosheng Zhang[1,2,3], Kehai Chen[4], Rui Wang[4,\*], Masao Utiyama[4], Eiichiro Sumita[4], Zuchao Li[1,2,3] , Hai Zhao[1,2,3,\*]**
[1]Department of Computer Science and Engineering, Shanghai Jiao Tong University
[2]Key Laboratory of Shanghai Education Commission for Intelligent Interaction
and Cognitive Engineering, Shanghai Jiao Tong University, Shanghai, China
[3]MoE Key Lab of Artificial Intelligence, AI Institute, Shanghai Jiao Tong University
[4]National Institute of Information and Communications Technology (NICT), Kyoto, Japan
`zhangzs@sjtu.edu.cn, charlee@sjtu.edu.cn, zhaohai@cs.sjtu.edu.cn,`
`{khchen, wangrui, mutiyama, eiichiro.sumita}@nict.go.jp`

## Abstract

Though visual information has been introduced for enhancing neural machine translation (NMT), its effectiveness strongly relies on the availability of large amounts of bilingual parallel sentence pairs with manual image annotations. In this paper, we present a universal visual representation learned over the monolingual corpora with image annotations, which overcomes the lack of large-scale bilingual sentence-image pairs, thereby extending image applicability in NMT. In detail, a group of images with similar topics to the source sentence will be retrieved from a light topic-image lookup table learned over the existing sentence-image pairs, and then is encoded as image representations by a pre-trained ResNet. An attention layer with a gated weighting is to fuse the visual information and text information as input to the decoder for predicting target translations. In particular, the proposed method enables the visual information to be integrated into large-scale text-only NMT in addition to the multimodal NMT. Experiments on four widely used translation datasets, including the WMT'16 English-to-Romanian, WMT'14 English-to-German, WMT'14 English-to-French, and *Multi30K*, show that the proposed approach achieves significant improvements over strong baselines.

## 1 Introduction

Visual information has been introduced for neural machine translation in some previous studies (NMT) (Specia et al., 2016; Elliott et al., 2017; Barrault et al., 2018; Ive et al., 2019) though the contribution of images is still an open question (Elliott, 2018; Caglayan et al., 2019). Typically, each bilingual (or multilingual) parallel sentence pair is annotated manually by one image describing the content of this sentence pair. The bilingual parallel corpora with manual image annotations are used to train a multimodal NMT model by an end-to-end framework, and results are reported on a specific data set, *Multi30K* (Calixto & Liu, 2017; Calixto et al., 2017).

One strong point of the multimodal NMT model is the ability to use visual information to improve the quality of the target translation. However, the effectiveness heavily relies on the availability of bilingual parallel sentence pairs with manual image annotations, which hinders the image applicability to the NMT. As a result, the visual information is only applied to the translation task over a small and specific multimodal data set *Multi30K* (Elliott et al., 2016), but not to large-scale text-only NMT (Bahdanau et al., 2014; Gehring et al., 2017; Vaswani et al., 2017) and low-resource

---

[\*]Corresponding author. Zhuosheng Zhang and Zuchao Li were internship research fellows at NICT when this work was done. Hai Zhao was partially supported by Key Projects of National Natural Science Foundation of China (No. U1836222 and No. 61733011). Rui Wang was partially supported by JSPS grant-in-aid for early-career scientists (19K20354): "Unsupervised Neural Machine Translation in Universal Scenarios" and NICT tenure-track researcher startup fund "Toward Intelligent Machine Translation."

text-only NMT (Fadaee et al., 2017; Lample et al., 2018; Ma et al., 2019; Zhou et al., 2019). In addition, because of the high cost of annotation, the content of one bilingual parallel sentence pair is only represented by a single image, which is weak in capturing the diversity of visual information. The current situation of introducing visual information results in a bottleneck in the multimodal NMT and is not feasible for text-only NMT and low-resource NMT.

In this paper, we present a universal visual representation (VR) method[1] relying only on image-monolingual annotations instead of the existing approach that depends on image-bilingual annotations, thus breaking the bottleneck of using visual information in NMT. In detail, we transform the existing sentence-image pairs into a topic-image lookup table from a small-scale multimodal data set *Multi30K*. During the training and decoding process, a group of images with a similar topic to the source sentence will be retrieved from the topic-image lookup table learned by the term frequency-inverse document frequency, and thus is encoded as image representations by a pre-trained ResNet (He et al., 2016). A simple and effective attention layer is then designed to fuse the image representations and the original source sentence representations as input to the decoder for predicting target translations. In particular, the proposed approach can be easily integrated into the text-only NMT model without annotating large-scale bilingual parallel corpora. The proposed method was evaluated on four widely-used translation datasets, including the WMT'16 English-to-Romanian, WMT'14 English-to-German, WMT'14 English-to-French, and *Multi30K* which are standard corpora for NMT and multimodal machine translation (MMT) evaluation. Experiments and analyses show effectiveness. In summary, our contributions are primarily three-fold:

1. We present a universal visual representation method that overcomes the shortcomings of the bilingual (or multilingual) parallel data with manual image annotations for MMT.

2. The proposed method enables the text-only NMT to use the multimodality of visual information without annotating the existing large scale bilingual parallel data.

3. Experiments on different scales of translation tasks verified the effectiveness and generality of the proposed approach.

## 2    RELATED WORK

Building fine-grained representation with extra knowledge is an essential topic in language modeling (Li et al., 2020a;b; Zhang et al., 2020b;a), among which adopting visual modality could potentially benefit the machine with a more comprehensive perception of the real world. Inspired by the studies on the image description generation (IDG) task (Mao et al., 2014; Elliott et al., 2015; Venugopalan et al., 2015; Xu et al., 2015), a new shared translation task for multimodal machine translation was addressed by the machine translation community (Specia et al., 2016). In particular, the released dataset *Multi30K* (Elliott et al., 2016) includes 29,000 multilingual (English, German, and French) parallel sentence pairs with image annotations (Elliott et al., 2017; Barrault et al., 2018). Subsequently, there has been a rise in the number of studies (Caglayan et al., 2016; 2017; Calixto et al., 2016; Huang et al., 2016; Libovickỳ & Helcl, 2017; Helcl et al., 2018). For example, Calixto et al. (2017) proposed a doubly-attentive multimodal NMT model to incorporate spatial-visual features, improving translation performance. Compared with spatial-visual features, Calixto & Liu (2017) further incorporated global image features as words in the source sentence and to enhance the encoder or decoder hidden state. In contrast, some recent studies indicated that the visual modality is either unnecessary (Zhang et al., 2017) or only marginally beneficial (Grönroos et al., 2018). More recently, Ive et al. (2019) showed that visual information is only needed in particular cases, such as for ambiguous words where the textual context is not sufficient.

However, these approaches only center around a small and specific *Multi30K* data set to build a multimodal NMT model, which hinders image applicability to NMT. The reason would be the high cost of image annotations, resulting potentially in the image information not being adequately discovered. We believe that the capacity of MMT has not yet been excavated sufficiently, and there is still a long way to go before the potential of MMT is fully discovered. In this work, we seek to break this constraint and enable visual information to benefit NMT, especially text-only NMT.

---

[1]The code is publicly available at `https://github.com/cooelf/UVR-NMT`.

## 3 UNIVERSAL VISUAL RETRIEVAL

---

**Algorithm 1** Topic-image Lookup Table Conversion Algorithm

---
**Require:** Input sentences, $S = \{X_1, X_2, \ldots X_I\}$ and paired images $E = \{e_1, e_2, \ldots, e_I\}$
**Ensure:** Topic-image lookup table $\mathcal{Q}$ where each word is associated with a group of images
  1: Obtain the TF-IDF dictionary $\mathcal{F}$ = TF-IDF($S$)
  2: Transform sentence-image pair to topic-image lookup table $\mathcal{Q}$ = LookUp($S, E, \mathcal{F}$)
  3: **procedure** TF-IDF($S$)
  4:     **for** each sentence in $S$ **do**
  5:         Filter stop-words in the sentence
  6:         Calculate the TF-IDF weight for each word
  7:     **end for**
  8:     **return** TF-IDF dictionary $\mathcal{F}$
  9: **end procedure**
 10: **procedure** LOOKUP($S, E, \mathcal{F}$)
 11:     **for** For each pair $\{T_i, e_i\} \in \texttt{zip}\{S, E\}$ **do**
 12:         Rank and pick out the top-$w$ "topic" words in the sentence according to the TF-IDF
    score in the dictionary $\mathcal{F}$, and each sentence is reformed as $T = \{t_1, t_2, \ldots, t_w\}$
 13:         Pair the $w$ words with the corresponding image $e_i$
 14:         **for** For each word $t_j$ in $T$ **do**
 15:             **if** $e_i$ not in $\mathcal{Q}[t_j]$ **then**
 16:                 Add $e_j$ to the corresponding image set $\mathcal{Q}[t_j]$ for word $t_j$
 17:             **end if**
 18:         **end for**
 19:     **end for**
 20:     **return** Topic-image lookup table $\mathcal{Q}$
 21: **end procedure**

---

In this section, we will introduce the proposed universal visual representation method. Generally, the default input setting of the MMT is a sentence-image pair. Our basic intuition is to transform the existing sentence-image pairs into topic-image lookup table[2], which assumes the topic words in a sentence should be relevant to the paired image. Consequently, a sentence can possess a group of images by retrieving the topic-image lookup table.

**Topic-image Lookup Table Conversion**   To focus on the major part of the sentence and suppress the noise such as stopwords and low-frequency words, we design a filtering method to extract the "topic" words of the sentence through the term frequency-inverse document frequency (TF-IDF)[3] inspired by Chen et al. (2019). Specifically, given an original input sentence $X = \{x_1, x_2, \ldots, x_I\}$ of length $I$ and its paired image $e$, $X$ is first filtered by a stopword list[4] and then the sentence is treated as a document $g$. We then compute TF-IDF $TI_{i,j}$ for each word $x_i$ in $g$,

$$TI_{i,j} = \frac{o_{i,j}}{\sum_k o_{k,j}} \times \log \frac{|G|}{1 + |j : x_i \in g|}, \tag{1}$$

where $o_{i,j}$ represents the number of occurrences of the word $x_i$ in the input sentence $g$, $|G|$ the total number of source language sentences in the training data, and $|j : x_i \in g|$ the number of source sentences including word $x_i$ in the training data. We then select the top-$w$ high TF-IDF words as the new image description $T = \{t_1, t_2, \ldots, t_w\}$ for the input sentence $X$. After preprocessing, each filtered sentence $T$ is paired with an image $e$, and each word $t_i \in T$ is regarded as the topic word for image $e$. After processing the whole corpus (i.e., *Multi30K*), we form a topic-image lookup table $\mathcal{Q}$ as described in Algorithm 1, in which each topic word $t_i$ would be paired with dozens of images.

**Image Retrieval**   For the input sentence, we first obtain its topic words according to the text preprocessing method described above. Then we retrieve the associated images for each topic word

---

[2] We use the training set of the *Multi30K* dataset to build the topic-image lookup table.

[3] We describe our methods by regarding the processing unit as word though this method can also be applied to a subword-based sentence for which the subword is considered to be the processing unit.

[4] https://github.com/stopwords-iso/stopwords-en

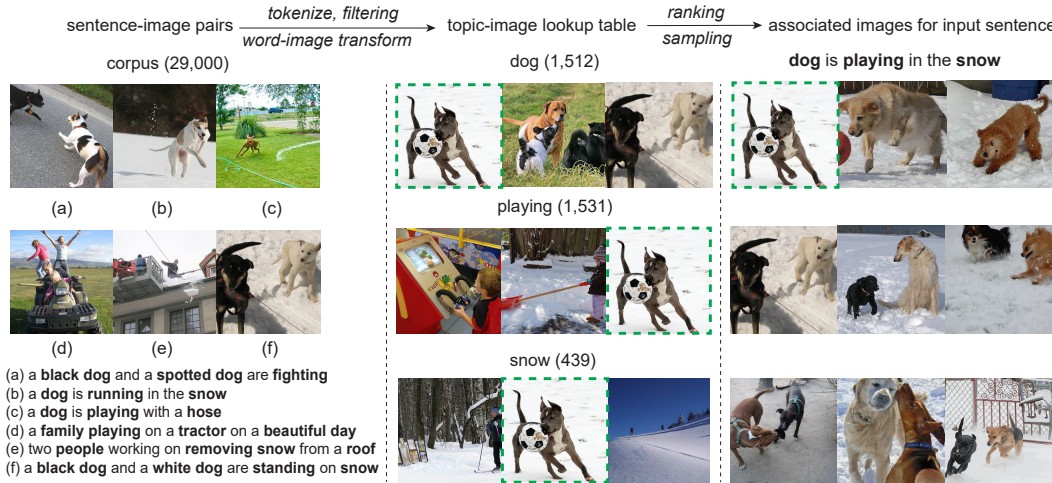

Figure 1: Illustration of the proposed visual retrieval.

from the lookup table $\mathcal{Q}$ and group all the retrieved images together to form an image list $\mathcal{G}$. We observe that an image might be associated with multiple topic words so that it would occur multiple times in the list $\mathcal{G}$. Therefore, we sort the images according to the frequency of occurrences in $\mathcal{G}$ to maintain the total number of images for each sentence at $m$.

Figure 1 illustrates the retrieval process[5]. In the left block, we show six examples of sentence-image pairs in which the topic words are in boldface. Then we process the corpus using the topic-image transformation method demonstrated above and obtain the topic-image lookup table. For example, the word $dog$ is associated with 1,512 images. For an input source sentence, we obtain the topic words (in boldface) using the same preprocessing. Then we retrieve the corresponding images from the lookup table for each topic word. Now we have a list of images, and some images appear multiple times as they have various topics (like the boxed image in Figure 1). So we sort the retrieved image list by the count of occurrence to pick out the top-$m$ images that cover the most topics of the sentence.

At test time, the process of getting images is done using the image lookup table built by the training set, so we do not need to use the images from the dev and test sets in *Multi30K* dataset[6]. Intuitively, we do not strictly require the manual alignment of the word (or concept) and image, but rely on the co-occurrence of topic word and image, which is simpler and more general. In this way, we call our method as universal visual retrieval.

## 4 NMT WITH UNIVERSAL VISUAL REPRESENTATION

In this section, we introduce the proposed universal visual representation (VR) method for NMT. The overview of the framework of our proposed method is shown in Figure 2.

### 4.1 SOURCE REPRESENTATION FOR NEURAL MACHINE TRANSLATION

In the state-of-the-art Transformer-based NMT (Vaswani et al., 2017), source information is encoded as source representation by an SAN-based encoder with multiple layers. Specifically, the encoder is composed of a stack of $L$ identical layers, each of which includes two sub-layers. The first sub-layer is a self-attention module, whereas the second is a position-wise, fully connected feed-forward network. A residual connection (He et al., 2016) is applied between the two sub-layers, and then

---

[5]More examples are provided in the Appendix A.1.

[6]The lookup table can be easily adapted to a wide range of other NLP tasks even without any paired image, and therefore opens our proposed model to generalization.

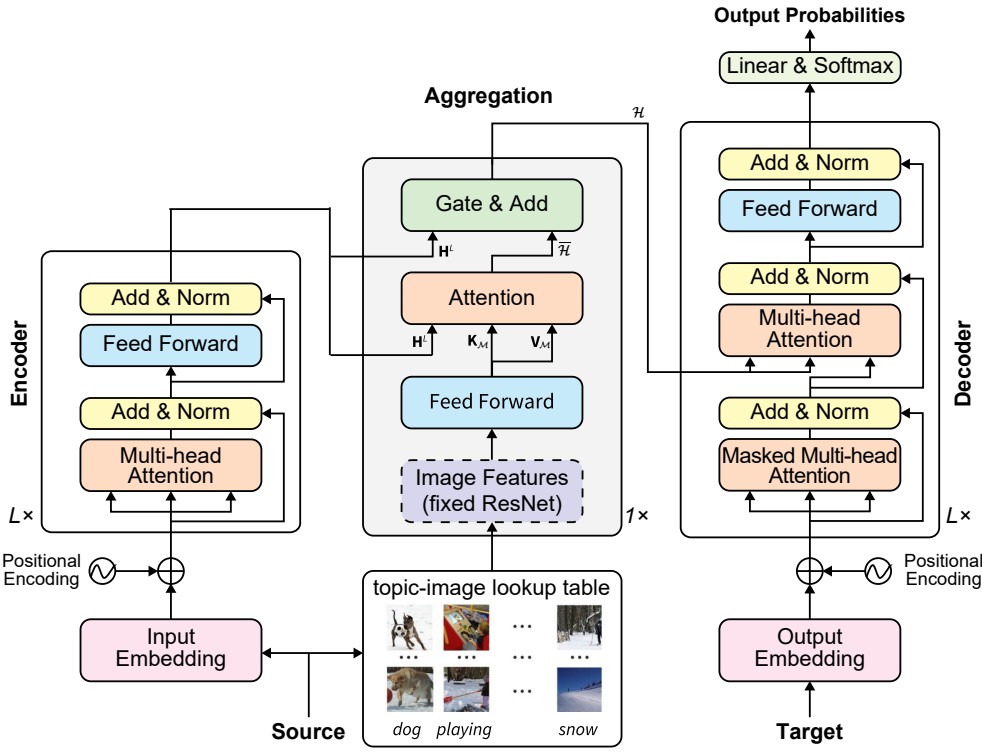

Figure 2: Overview of the framework of our proposed method.

a layer normalization (Ba et al., 2016) is performed. Formally, the stack of learning the source representation is organized as follows:

$$\overline{\mathbf{H}}^l = \text{LN}(\text{ATT}^l(\mathbf{Q}^{l-1}, \mathbf{K}^{l-1}, \mathbf{V}^{l-1}) + \mathbf{H}^{l-1}),$$
$$\mathbf{H}^l = \text{LN}(\text{FFN}^l(\overline{\mathbf{H}}^l) + \overline{\mathbf{H}}^l), \tag{2}$$

where $\text{ATT}^l(\cdot)$, $\text{LN}(\cdot)$, and $\text{FFN}^l(\cdot)$ are the attention module, layer normalization, and the feed-forward network for the $l$-th identical layer, respectively. $\{\mathbf{Q}^{l-1}, \mathbf{K}^{l-1}, \mathbf{V}^{l-1}\}$ are query, key, and value vectors that are transformed from the $(l$-1)-th layer $\mathbf{H}^{l-1}$. For example, $\{\mathbf{Q}^0, \mathbf{K}^0, \mathbf{V}^0\}$ are packed from the summation $\mathbf{H}^0$ of the positional embeddings and word embeddings. Finally, the output of the stack of $L$ identical layers $\mathbf{H}^L$ is the final source sentence representation.

## 4.2 AGGREGATION FOR TEXT AND IMAGE REPRESENTATIONS

After retrieval as described in Section 3, each original sentence $X = \{x_1, x_2, \ldots, x_I\}$ is paired with $m$ images $E = \{e_1, e_2, \ldots, e_m\}$ retrieved from the topic-image lookup table $\mathcal{Q}$. First, the source sentence $X = \{x_1, x_2, \ldots, x_I\}$ is fed into the encoder (Eq.2) to learn the source sentence representation $\mathbf{H}^L$. Second, the images $E = \{e_1, e_2, \ldots, e_m\}$ are the inputs to a pre-trained ResNet (He et al., 2016) followed by a feed forward layer to learn the source image representation $text{M} \in R^{m \times 2048}$. Then, we apply an attention mechanism[7] to append the image representation to the text representation:

$$\overline{\mathcal{H}} = \text{ATT}_{\mathcal{M}}(\mathbf{H}^L, \mathbf{K}_{\mathcal{M}}, \mathbf{V}_{\mathcal{M}}), \tag{3}$$

where $\{\mathbf{K}_{\mathcal{M}}, \mathbf{V}_{\mathcal{M}}\}$ are packed from the learned source image representation $\mathcal{M}$.

---

[7]We used single head here for simplicity.

Intuitively, NMT aims to produce a target word sequence with the same meaning as the source sentence rather than a group of images. In other words, the image information may play an auxiliary effect during the translation prediction. Therefore, we compute $\lambda \in [0, 1]$ to weight the expected importance of source image representation for each source word:

$$\lambda = \text{sigmoid}(\mathbf{W}_\lambda \overline{\mathcal{H}} + \mathbf{U}_\lambda \mathbf{H}^L), \qquad (4)$$

where $\mathbf{W}_\lambda$ and $\mathbf{U}_\lambda$ are model parameters. We then fuse $\mathbf{H}^L$ and $\overline{\mathcal{H}}$ to learn an effective source representation:

$$\mathcal{H} = \mathbf{H}^L + \lambda \overline{\mathcal{H}}. \qquad (5)$$

Finally, $\mathcal{H}$ is fed to the decoder to learn a dependent-time context vector for predicting target translation. Note that there is a single aggregation layer to fuse image and text information.

## 5 EXPERIMENTS

### 5.1 DATA

The proposed method was evaluated on four widely-used translation datasets, including WMT'16 English-to-Romanian (EN-RO), WMT'14 English-to-German (EN-DE), WMT'14 English-to-French (EN-DE), and *Multi30K* which are standard corpora for NMT and MMT evaluation.

1) For the EN-RO task, we experimented with the officially provided parallel corpus: Europarl v7 and SETIMES2 from WMT'16 with 0.6M sentence pairs. We used *newsdev2016* as the dev set and *newstest2016* as the test set.

2) For the EN-DE translation task, 4.43M bilingual sentence pairs of the WMT14 dataset were used as training data, including Common Crawl, News Commentary, and Europarl v7. The *newstest2013* and *newstest2014* datasets were used as the dev set and test set, respectively.

3) For the EN-FR translation task, 36M bilingual sentence pairs from the WMT14 dataset were used as training data. *Newstest12* and *newstest13* were combined for validation and *newstest14* was used as the test set, following the setting of Gehring et al. (2017).

4) The *Multi30K* dataset contains 29K English→{German, French} parallel sentence pairs with visual annotations. The 1,014 English→{German, French} sentence pairs visual annotations are as dev set. The test sets are test2016 and test2017 with 1,000 pairs for each.

### 5.2 SYSTEM SETTING

**Image Retrieval Implementation**  We used 29,000 sentence-image pairs from *Multi30K* to build the topic-image lookup table. We segmented the sentences using the same BPE vocabulary as that for each source language. We selected top-8 ($w = 8$) high TF-IDF words, and the default number of images $m$ was set 5. The detailed case study is shown in Section 6.2. After preprocessing, we had about 3K topic words, associated with a total of 10K images for retrieval. Image features were extracted from the averaged pooled features of a pre-trained ResNet50 CNN (He et al., 2016). This led to feature maps $V \in R^{2048}$.

**Baseline**  Our baseline was text-only Transformer (Vaswani et al., 2017). We used six layers for the encoder and the decoder. The number of dimensions of all input and output layers was set to 512 and 1024 for *base* and *big* models. The inner feed-forward neural network layer was set to 2048. The heads of all multi-head modules were set to eight in both encoder and decoder layers. For the *Multi30K* dataset, we further evaluated a multimodal baseline (denoted as MMT) where each source sentence was paired with an original image. The other settings were the same as our proposed model.

**Model Implementation**  The byte pair encoding algorithm was adopted, and the size of the vocabulary was set to 40,000. In each training batch, a set of sentence pairs contained approximately 4096×4 source tokens and 4096×4 target tokens. During training, the value of label smoothing was set to 0.1, and the attention dropout and residual dropout were $p = 0.1$. We used Adam optimizer (Kingma & Ba, 2014) to tune the parameters of the model. The learning rate was varied

| System | Architecture | EN-RO | | EN-DE | | EN-FR | |
|---|---|---|---|---|---|---|---|
| | | BLEU | #Param | BLEU | #Param | BLEU | #Param |
| *Existing NMT systems* | | | | | | | |
| Vaswani et al. (2017) | Trans. (base) | N/A | N/A | 27.3 | N/A | 38.1 | N/A |
| | Trans. (big) | N/A | N/A | 28.4 | N/A | 41.0 | N/A |
| Lee et al. (2018) | Trans. (base) | 32.40 | N/A | 24.57 | N/A | N/A | N/A |
| *Our NMT systems* | | | | | | | |
| This work | Trans. (base) | 32.66 | 61.54M | 27.31 | 63.44M | 38.52 | 63.83M |
| | +VR | **33.78++** | 63.04M | **28.14++** | 64.94M | **39.64++** | 65.33M |
| | Trans. (big) | 33.85 | 207.02M | 28.45 | 210.88M | 41.10 | 211.66M |
| | +VR | **34.46+** | 211.02M | **29.14++** | 214.89M | **41.83+** | 215.66M |

Table 1: Results on EN-RO, EN-DE, and EN-FR for the NMT tasks. Trans. is short for transformer. N/A denotes that those numbers are not reported in the corresponding literature. "++/+" after the BLEU score indicate that the proposed method was significantly better than the corresponding baseline Transformer (base or big) at significance level $p < 0.01/0.05$.

under a warm-up strategy with 8,000 steps. For evaluation, we validated the model with an interval of 1,000 batches on the dev set. For the *Multi30K* dataset, we trained the model up to 10,000 steps, and the training was early-stopped if *dev* set BLEU score did not improve for ten epochs. For the EN-DE, EN-RO, and EN-FR tasks, following the training of 200,000 batches, the model with the highest BLEU score of the dev set was selected to evaluate the test sets. During the decoding, the beam size was set to five. All models were trained and evaluated on a single V100 GPU. Multi-bleu.perl[8] was used to compute case-sensitive 4-gram BLEU scores for all test sets. The signtest (Collins et al., 2005) is a standard statistical-significance test. In addition, we followed the model configurations of Vaswani et al. (2017) to train Big models for WMT EN-RO, EN-DE, and EN-FR translation tasks. All experiments were conducted with *fairseq*[9] (Ott et al., 2019). The analysis in Section 6 is conducted on base models.

## 5.3 RESULTS

Table 1 shows the results for the WMT'14 EN-DE, EN-FR, and WMT'16 EN-RO translation tasks. Our implemented Transformer (base/big) models showed similar BLEU scores with the original Transformer (Vaswani et al., 2017), ensuring that the proposed method can be evaluated over strong baseline NMT systems. As seen, the proposed +VR significantly outperformed the baseline Transformer (base), demonstrating the effectiveness of modeling visual information for text-only NMT. In particular, the effectiveness was adapted to the translation tasks of the three language pairs, which have different scales of training data, verifying that the proposed approach is a universal method for improving translation performance.

Our method introduced only 1.5M and 4.0M parameters for the base and big transformers, respectively. The number is less than 3% of the baseline parameters as we used the fixed image embeddings from the pre-trained ResNet feature extractor. Besides, the training time was basically the same as the baseline model (Section 6.4).

In addition, the proposed method was also evaluated for MMT on the multimodal dataset, *Multi30K*. Results in Table 2 show that our model also outperformed the transformer baseline. Compared with the results in text-only NMT, we find that the image presentation gave marginal contribution, which was consistent with the findings in previous work (Zhang et al., 2017; Grönroos et al., 2018; Caglayan et al., 2019). The most plausible reason might be that the sentences in *Multi30K* are so simple, short, and repetitive that the source text is sufficient to perform the translation (Caglayan et al., 2019; Ive et al., 2019). This verifies our assumption of the current bottleneck of MMT due to the limitation of *Multi30K* and shows the necessity of our new setting of transferring multimodality into more standard and mature text-only NMT tasks.

---

[8] https://github.com/moses-smt/mosesdecoder/tree/RELEASE-4.0/scripts/generic/multi-bleu.perl
[9] https://github.com/pytorch/fairseq

| System | Architecture | EN-DE | | | EN-FR | | |
|---|---|---|---|---|---|---|---|
| | | Test2016 | Test2017 | #Param | Test2016 | Test2017 | #Param |
| *Existing NMT systems* | | | | | | | |
| Calixto et al. (2017) | RNN | 33.7 | N/A | N/A | N/A | N/A | N/A |
| Elliott et al. (2017) | RNN | N/A | 19.3 | N/A | N/A | 44.3 | N/A |
| Elliott & Kádár (2017) | Imagination | 36.8 | N/A | N/A | N/A | N/A | N/A |
| Ive et al. (2019) | Trans. (big) | 36.4 | N/A | N/A | 59.0 | N/A | N/A |
| | Del | 38.0 | N/A | N/A | 60.1 | N/A | N/A |
| *Our MMT systems* | | | | | | | |
| This work | MMT. (base) | 35.09 | 27.10 | 50.72M | 57.40 | 48.02 | 50.65M |
| | MMT. (big) | 35.60 | 28.02 | 190.58M | 57.87 | 49.63 | 190.43M |
| | Trans. (base) | 35.59 | 26.31 | 49.15M | 57.88 | 48.55 | 49.07M |
| | **+VR** | **35.72** | **26.87** | 50.72M | **58.32** | **48.69** | 50.65M |
| | Trans. (big) | 36.86 | 27.62 | 186.38M | 56.97 | 48.17 | 186.23M |
| | **+VR** | **36.94** | **28.63** | 190.58M | **57.53** | **48.46** | 190.43M |

Table 2: Results from the test2016 and test2017 for the MMT task. Del denotes the deliberation network in (Ive et al., 2019). Elliott et al. (2017) is the official baseline (text-only NMT) on WMT17-Multi30K 2017 test data. Trans. is short for transformer and MMT is the multimodal baseline described in Section 5.2. Because we used the same model for test2016 and test2017 evaluation, the numbers of parameters are the same.

# 6 ANALYSIS

## 6.1 WHY DOES THE LOOKUP TABLE WORK

The contribution of the lookup table could be two folds: 1) the content connection of the sentences and images; 2) the topic-aware co-occurrence of similar images and sentences. There are cases when paired images are not accurately related to the given sentence. A simple solution is to set a threshold heuristically for the TF-IDF retrieval to filter out the "improper" images. However, we maintain the specific number of the images in this work because of the second potential benefits of the co-occurrence, by taking images as diverse topic information. According to Distributional Hypothesis (Harris, 1954), which states that *words that occur in similar contexts tend to have similar meanings*, we are inspired to extend the concept in the multimodal world, *the sentences with similar meanings would be likely to pair with similar even the same images*. Therefore, the consistent images (with a related topic) could play the role of topic or type hints for similar sentence modeling.

This is also very similar to the idea of word embedding by taking each image as a "word". Because we use the average pooled output of ResNet, each image is represented as a 2400-d vector. For all the 29,000 images, we have an embedding layer with size (29000, 2400). The "content" of the image is regarded as the embedding initialization. It indeed makes effects, but the capacity of the neural network is not up to it. In contrast, the mapping from text word to the index in the word embedding is critical. Similarly, the mapping of sentence to image in image embedding would be essential, i.e., the similar sentences (with the same topic words) tend to map the same or similar image.

To verify the hypotheses, we replace our ResNet features with 1) *Shuffle*: shuffle the image features but keep the lookup table; 2) *Random Init*: randomly initialize the image embedding but keep the lookup table; 3) *Random Mapping*: randomly retrieve unrelated images. The BLEU scores are on EN-RO are 33.53, 33,28, 32.14, respectively. The results of 1-2 are close to the proposed VR (33.78) and outperform the baseline (32.66), which shows that the content of images would not be very important. The ablation 3) gives a lower result, which verifies the necessity of the mapping, especially the topic relationship.

## 6.2 INFLUENCE OF THE NUMBER OF IMAGES

To evaluate the influence of the number of paired images $m$, we constrained $m$ in {0, 1, 3, 5, 7, 9, 15, 20, 30} for experiments on the EN-RO test set, as shown in Figure 4. When $m = 0$, the model is the baseline NMT model, whose BLEU score was lower than all the models with images. As the number of images increases, the BLEU score also increased at the beginning (from 32.66 to 33.78)

and then slightly decreased when $m$ exceeds 5. The reason might be that too many images for a sentence would have a higher chance of noise. Therefore, we set $m = 5$ in our models.

The number of sentence-image pairs to create the lookup table could also make effects. We randomly split the pairs of Multi30K into the proportion in [0.1, 0.3, 0.5, 0.7, 0.9], the corresponding BLEU scores for EN-RO are [33.07, 33.44, 34.01, 34.06, 33.80]. Furthermore, we also evaluate the performance by adding external sentence-pairs from the training set of MS COCO image caption dataset (Lin et al., 2014). The BLEU scores are 33.55 and 33.71, respectively, for COCO only and Multi30K+COCO. These results indicate that a modest number of pairs would be beneficial.

### 6.3 THE INFLUENCE OF GATING WEIGHT $\lambda$

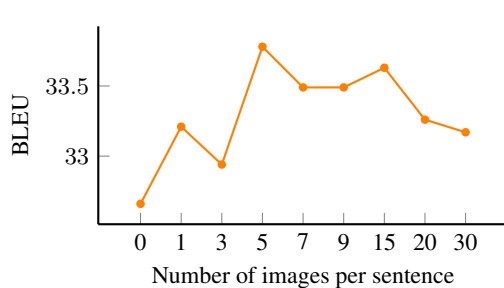

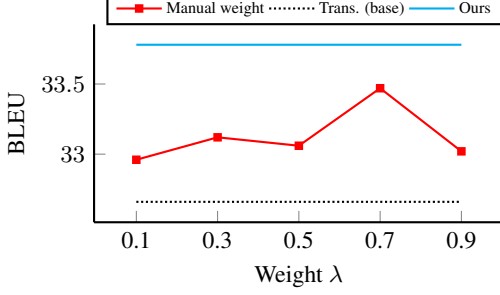

Figure 4: Influence of the number of images on the BLEU score.

Figure 5: Quantitative study of the gating weight $\lambda$.

In our model, the weight $\lambda$ of the gated aggregation method was learned automatically to measure the importance of the visual information. We compared by manually setting the weight $\lambda$ into scalar values in {0.1, 0.3, 0.5, 0.7, 0.9} for experiments on the EN-RO test set. Figure 5 shows that all models with manual $\lambda$ outperformed the baseline Trans. (base), indicating the effectiveness of image information. In contrast, they were inferior to the performance of our model. This means that the degree of dependency for image information varies for each source sentence, indicating the necessity of automatically learning the gating weights of image representations.

### 6.4 EXTRA COMPUTATION TIME

There are mainly two extra computation costs using our method, including 1) obtaining image data for sentences and 2) learning image representations, which are negligible compared with training an NMT model. The time of obtaining image data for MT sentences for the EN-RO dataset is less than 1 minute using GPU. The lookup table is formed as the mapping of token (only topic words) index to image id. Then, the retrieval method is applied as the tensor indexing from the sentence token indices (only topic words) to image ids, which is the same as the procedure of word embedding. The retrieved image ids are then sorted by frequency. Learning image representations takes about 2 minutes for all the 29,000 images in Multi30K using 6G GPU memory for feature extraction and eight threads of CPU for transforming images. The extracted features are formed as the "image embedding layer" with the size of (29000, 2400) for quick access in the neural network.

## 7 CONCLUSION

This work presents a universal visual representation method for neural machine translation relying on monolingual image annotations, which breaks the restraint of heavy dependency on bilingual sentence-image pairs in the current multimodal NMT setting. In particular, this method enables visual information to be applied to large-scale text-only NMT through a topic-image lookup. We hope this work sheds some light on future MMT research. In the future, we will try to adopt the proposed method for other tasks.

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

## A    APPENDIX

### A.1    EXAMPLES OF RETRIEVED IMAGES

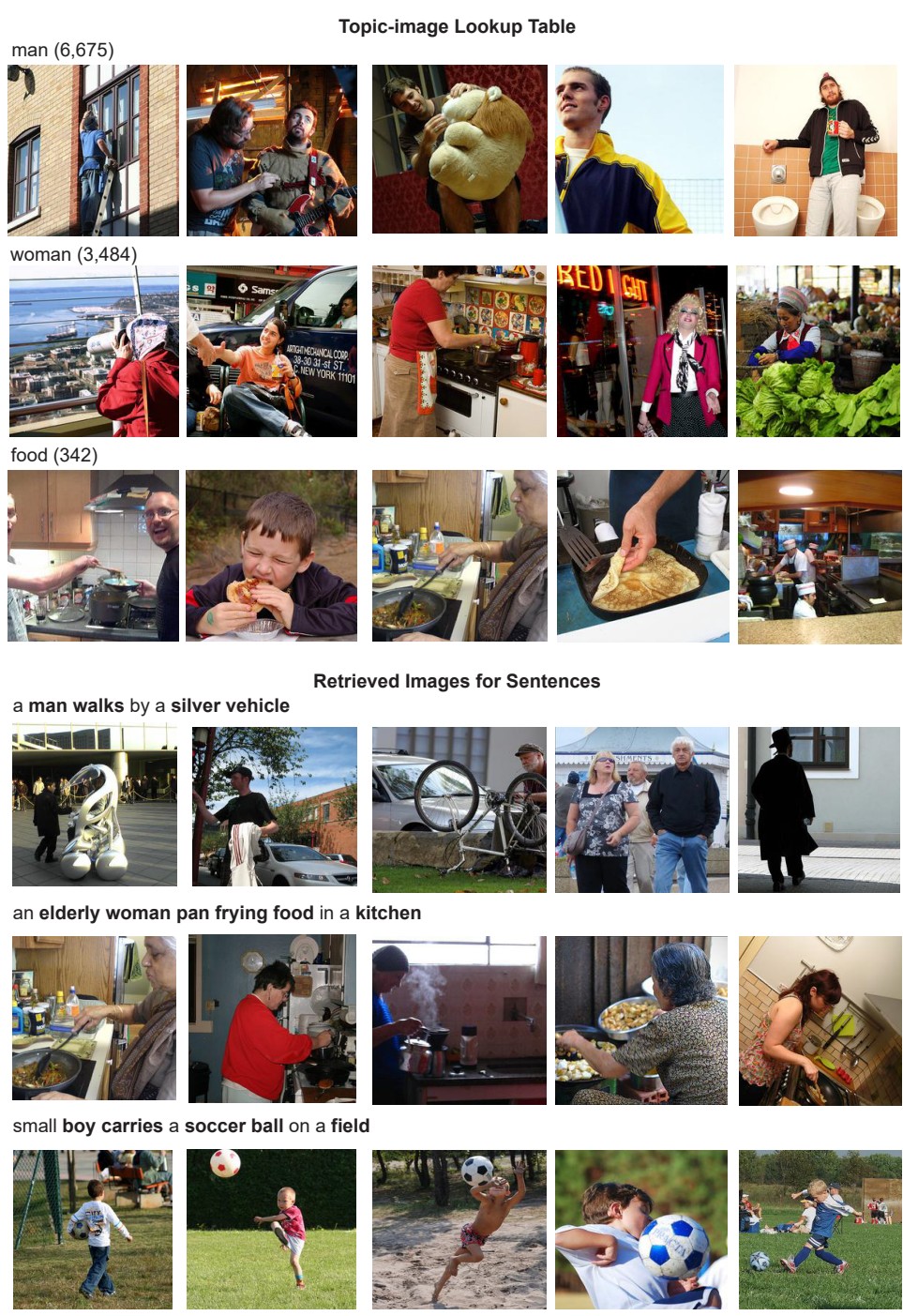

Figure 5: Examples of the topic-image lookup table and retrieved images for sentences in Multi30K dataset. We only show six images for each topic or sentence for instance. The topics in each sentence are in boldface.

**Retrieved Images for Sentences (WMT)**

The old system of **private** arbitration **courts** is off the **table**

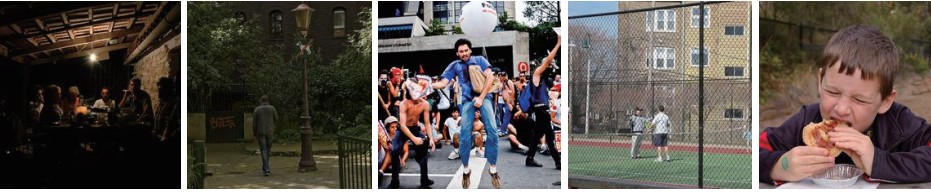

This **issue** is **shaping** as a potential early rift with the **business community**

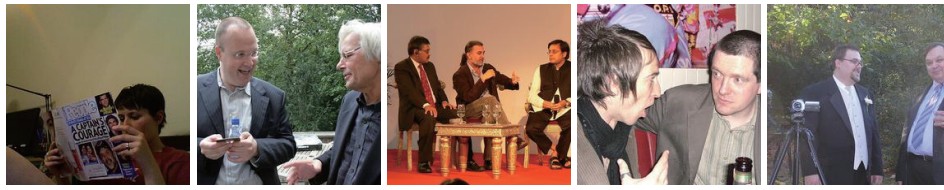

He said he then **heard** his **friend** , Hamza **calling** to him

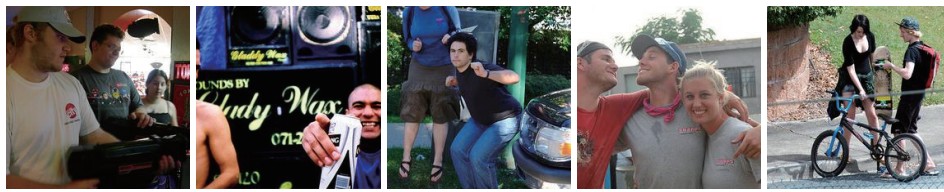

The **red flag** has been risen

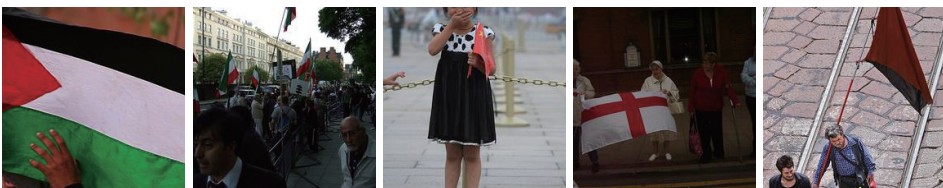

The **character attempts** to **pass human** smugglers and then **border police** on his way to a refugee **centre** in the European Union .

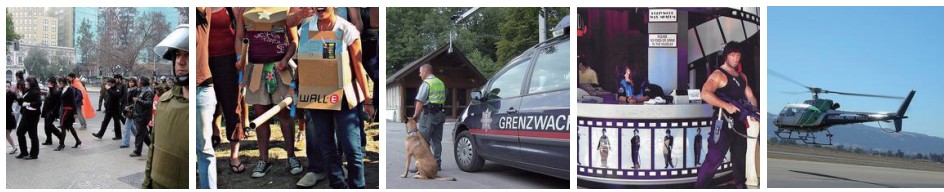

Figure 6: Examples of the retrieved images for sentences in WMT datasets. We only show six images for each sentence for instance. The topics in each sentence are in boldface.

