# OpenReview forum: "Neural Machine Translation with Universal Visual Representation"
_ICLR.cc/2020/Conference — Accept (Spotlight)_

### Official Review · AnonReviewer1 · 2019-10-23
**Official Blind Review #1**

**Rating:** 6

**Review:**

The authors propose to augment NMT with a grounded inventory of images.  The intuition is clear and the premise is very tempting.  The key architectural choice is to allow the transformer to use language embeddings to attend into a topic-image lookup table.  The proportion is learned to balance how much signal comes from each source.    Figure 4, attempts to investigate the importance of this sharing and its effects on performance.

While reviewing this paper I went back and read the EN-DE evaluation data for the last few years trying to see how often I could reason that images would help and I came up severely lacking.  For example, "The old system of private arbitration courts is off the table" from DE-EN 2016 Dev doesn't seem like it should benefit from this architecture.  It's then hard for me to square that with the +VR gains seen throughout this work on non-grounded datasets.  I trust that the authors did in fact achieve these results but I cannot figure out how or why.  This is all further confused by the semantic topics used for clustering the images which ignores stop words and therefore spatial relations or any grammatical nuances.

In contrast, it does make sense that Multi30K would benefit from this architecture.  As a minor note, were different feature extractors compared? The recent flurry of papers on multimodal transformers indicate that deeper resnet stacks correspond to improved downstream performance.  Is that also true in this domain?

**Experience Assessment:**

I have published one or two papers in this area.

**Review Assessment: Checking Correctness Of Derivations And Theory:**

N/A

**Review Assessment: Checking Correctness Of Experiments:**

I assessed the sensibility of the experiments.

**Review Assessment: Thoroughness In Paper Reading:**

N/A

---

> ### Author Response · Authors · 2019-11-09
> **Response to Reviewer #1**
>
> Thanks for your insightful comments.
>
> 1. How or why is the benefit.
> This comment is insightful and we also considered about it. Intuitively, we would easily fall into the connections between each sentence and image. However, it is nearly impossible to pair sentence with images with completely the same meaning all the time. According to our investigation, we conclude that the major contribution would be more effective contextualized sentence encoding for better representation from the visual clue combination instead of single image enhancement for encoding each individual sentence or word.
>
> According to Distributional Hypothesis (Harris et al., 1954) which states that “words that occur in similar contexts tend to have similar meanings”, we are inspired to extend the concept in multimodal world, “the sentences with similar meanings would be likely to pair with similar even the same images”, where the consistent images (with similar topic) could play the role of topic or type clues for similar sentence modeling. For your example, the topic words are {private, courts, table}, which can be paired with relevant images and other sentences with the same (similar) topics will be paired with the same (similar) group of images.
>
> This is also very similar to the idea of word embedding by taking each image as a “word”. Because we use the average pooled output of ResNet, each image is represented as 2400d vector. For all the 29,000 images, we have an embedding layer with size (29000, 2400). The “content” of the image is just like the embedding initialization. It indeed makes effects, but the capacity of the neural network is not up to it. In contrast, the mapping from text word to the index in the word embedding is critical. Similarly, the mapping of sentence to image in image embedding would be essential, i.e., the similar sentences (with the same topic words) tend to map the similar images.
>
> To verify the hypothesis, we shuffle the image embeddings but keep the lookup table, to only exchange the features of each image but maintain the sentence-image mapping. Unsurprisingly, the BLEU score (EN-RO) is 33.53, which is very close to the reported one (33.78). In addition, we randomly initialize the image embedding instead of ResNet, the result is 33.28. In comparison, if we randomly retrieve unrelated images to break the lookup, the result is 32.14. These results verify the necessity of the lookup table. We have added a detailed discussion in the paper (please see Analysis 6.1).
>
> We believe this finding would be suggestive for the future research since most previous work focused on the content of the image itself. As a different research line, we highlight the consistency among the mono-modality to bridge the gap of language and image modeling.
>
> 2. Why stop words are ignored.
> According to the explanation above, we think the spatial relations or grammatical nuances would not be so important in this task if we take the images as topic guidance. Ignoring the stopwords can help us get rid of the disturbance of unnecessary high-frequency words (such as function words) being the topic, as the standard practice for TF-IDF topic extraction.
>
> 3. Comparison of different feature extractors.
> Yes. We compared with ResNet101 and ResNet152 on EN-RO. The BLEU scores are 33.63 and 33.87. It seems deeper ResNet indeed gives better results but the difference is not very significant.

---

### Official Review · AnonReviewer2 · 2019-10-23
**Official Blind Review #2**

**Rating:** 8

**Review:**

This paper provides an approach to use visual information to improve text only neural machine translation systems. The approach creates a "topic word to images" map using an existing image aligned translation corpora. Given a source sentence, the model extracts relevant images, extracts their Resnet features and fuses them with the features generated from the word sequence. The decoder uses these fused representation to generate the target sentence. Overall, I like the approach, seems like it can be easily augmented to existing NMT systems.

One of the claims of the paper was to be able to use monolingual image aligned data. However image captioning datasets are not mentioned. It would make sense to use image captioning data to create the image lookup. Also, what will be the performance of a standard image captioning system on the task ? I believe it will not be great, but I think for completeness, you should add such a baseline.

Minor comments:
1. What is M in Algorithm 1 ?
2. First paragraph in related work is very unrelated to the current subject, please remove.


**Experience Assessment:**

I have published one or two papers in this area.

**Review Assessment: Checking Correctness Of Derivations And Theory:**

I assessed the sensibility of the derivations and theory.

**Review Assessment: Checking Correctness Of Experiments:**

I carefully checked the experiments.

**Review Assessment: Thoroughness In Paper Reading:**

I read the paper thoroughly.

---

> ### Author Response · Authors · 2019-11-09
> **Response to Reviewer #2**
>
> Thanks for your constructive feedbacks! Please see our response below.
>
> 1. About image captioning.
>
> Yes. Image captioning dataset is absolutely available for creating the lookup table. As you suggest, we use MS COCO Image captioning dataset to learn a lookup table and apply it to the EN-RO translation task to do the quick evaluation. As a result, the BLEU score is (33.55), which is comparable to the current lookup table (33.78) based on Multi30K, and outperforms the Trans. (base) (32.66).
>
> Regarding the performance of the standard image captioning system, we train a caption model (Show, Attend, and Tell (Xu et al., 2015b)) with fine-tuned encoder (ResNet101) on the COCO dataset to encode the images. The result on EN-RO is 33.58.  We are a little bit uncertain if we have well understood this request because our task is text to text translation while image captioning is image to text. If not, we are glad to address further.
>
> 2. About the minor comments.
>
> (1)	This is typo. It is Q.
>
> (2)	Yes. We will remove it following your suggestion.

---

### Official Review · AnonReviewer3 · 2019-10-24
**Official Blind Review #3**

**Rating:** 6

**Review:**

Summary: This paper uses visual representation learned over monolingual corpora with image annotations, which overcomes the lack of large-scale bilingual sentence-image pairs for multimodal NMT. Their approach enables visual information to be integrated into large-scale text-only NMT. Experiments on four widely used translation datasets show that the proposed approach achieves significant improvements over strong baselines.

Strengths:
- This paper is well motivated and well written. I especially like how they use external paired sentence-image data from Multi30k to learn weak pairs for sentences in machine translation.
- Experimental results are convincing. I like how low-resource translation is included as a priority in their experiments.

Weaknesses:
- Do you have any explanations as to why the number of images, if too large, actually hurts translation performance? Is it because more images also leads to a higher chance of noisy images?
- It would be nice to have an experiment that varies the size of the external paired sentence-image dataset and tested the impact on performance.
- Please comment on the extra computation required for obtaining image data for MT sentences and for learning image representations.
- Why are there missing BLEU scores and the number of parameters in Table 1?

### Post rebuttal ###
Thank you for your detailed answers to my questions.

**Experience Assessment:**

I have read many papers in this area.

**Review Assessment: Checking Correctness Of Derivations And Theory:**

N/A

**Review Assessment: Checking Correctness Of Experiments:**

I carefully checked the experiments.

**Review Assessment: Thoroughness In Paper Reading:**

I read the paper thoroughly.

---

> ### Author Response · Authors · 2019-11-09
> **Response to Reviewer #3**
>
> Thanks so much for your constructive feedbacks. Please see our response below.
>
> 1. Influence of the number of images:
> Yes. The reason might be the higher chance of noise. It would be very important to provide a group of images that share similar patterns or topics. However, too many images for a sentence would have greater chance of noise.
>
> 2. Impact of paired sentence-image dataset:
> Yes. We add the external MS COCO image caption training set and evaluate on the EN-RO task for quick evaluation. The BLEU scores are 33.55 and 33.71 respectively for COCO only and Multi30K+COCO.
>
> In addition, we are also interested in the influence of the number of sentence-image pairs inspired by your suggestion. We randomly split the pairs of Multi30K into the proportion in [0.1, 0.3, 0.5, 0.7, 0.9], the corresponding BLEU scores are [33.07, 33.44, 34.01, 34.06, 33.80] respectively. These results indicate that a modest number of pairs would be beneficial.
>
> 3. The extra computation:
> The extra computation is negligible.
>
> The time of obtaining image data for MT sentences for EN-RO dataset, for example, is approximately less than 1 minute by tensor operation in GPU. The lookup table is formed as the mapping of token (only topic words) index to image id. Then, the retrieval method is applied as the tensor indexing from the sentence token (only topic words) index to image ids, which is the same as the procedure of word embedding. The retrieved image ids are then sorted by frequency.
>
> Learning image representations takes only about 2 minutes for all the 29,000 images in Multi30K using 6G GPU memory for feature extraction and 8 threads of CPU for transforming images. The extracted features are formed as the “image embedding layer” with the size of (29000, 2400) for quick accessing in neural network.
>
> 4. Missing BLEU scores & the number of parameters:
> Because those missing numbers (N/A) are not reported in the corresponding literature.

---

### Public Comment · ~Desmond_Elliott1 · 2019-10-01
**Comment about the benefits of visual information and the system comparisons in Table 2**

The introduction of your paper claims that "Visual information has been shown beneficial in neural machine translation (NMT) (Specia et al.,2016; Elliott et al., 2017; Barrault et al., 2018)." I think this is still an open question that is being debated in the literature, see Elliott (EMNLP 2018) and Caglayan et al. (NAACL 2019) for more details. A recent paper by Ive et al. ACL 2019 with a deliberation network claims that visual information is in fact beneficial but you did not include that model in your table of results.

Also, in Table 2, Elliott et al. (2017) is a not a system - it is a shared task overview paper. Elliott and Kádár (2017; http://aclweb.org/anthology/I17-1014) is a system that was evaluated on English-German multimodal translation. Perhaps this is a typo in the table, or you meant a different system that is decribed in the Elliott et al. WMT 2017 overview paper.

---

> ### Author Response · Authors · 2019-10-02
> **Response**
>
> Thanks for your interest and constructive comments!
>
> Response-to-comment-1. In the introduction, we intend to indicate that visual representation has been shown beneficial by some studies. As you mentioned, we agree that it is an open question, as we discussed in Section 2 and Section 5.3. We will clarify that “it is still an open question” in the introduction part and add more discussions accordingly in the later version.
>
> In previous literatures, visual information is primarily applied to the translation task over Multi30K dataset. Nevertheless, the conclusion about the benefit of the visual modality is still unclear. In this work, we propose to investigate the effectiveness in different and more universal scenarios. We are motivated to comprehensively explore and evaluate the potential of visual modality on more datasets that are in different scales. Our method relies only on image-monolingual annotations instead of the existing approach that depends on image-bilingual annotations, thus breaking the bottleneck of using visual information in NMT. Our experimental results and analysis verify the effectiveness of the visual representation.
>
> Based on the motivation above, we primarily focus on the evaluations of the stable improvements on the baselines (instead of SOTA comparisons), especially for the text-only NMT and low-resource NMT (Table 1) without manually-annotated text-image pairs. Therefore, we show the results of some public baseline models (including the transformer in Ive et al. (2019), instead of the deliberation network) for reference in Table 1-2, only to indicate that our implemented transformer models showed similar BLEU scores with other public reported transformers. As you mentioned, we will also add the SOTA performances in the revised version to show the effect of visual information in various scenarios.
>
> Response-to-comment-2. That is the official baseline (text-only NMT) on WMT17-Multi30K 2017 test data in Elliott et al. (2017). We will make it clear by noting in the table caption to avoid confusing. In addition, we will add the result of EN-DE in Elliott and Kádár (2017) for more comprehensive reference.

---

### Author Response · Authors · 2019-11-09
**Submission Update**

We thank all reviewers so much for the valuable comments on improving the quality of this work. We have updated the paper according to the feedback and our latest evaluations. The major revisions are marked in red for easy reading.

1)	We add a discussion (Analysis 6.1) about the contribution of the lookup table for the improved results. The comparisons of different feature extractors are also included in this section. Detailed demonstrations are in Appendix A.2.

2)	We add the discussion (Analysis 6.2) to demonstrate the effects of the number of sentence-image pairs including splitting the Multi30K and adding external MS COCO image caption datasets for comparisons.

3)	We add the discussion of external computation time in Appendix A.1.

4)	We add a page of more retrieved images for sentences in WMT datasets in Appendix A.4.

---

### Decision · Program_Chairs · 2019-12-19

**Decision:**

Accept (Spotlight)

**Comment:**

This paper proposes using visual representations learned in a monolingual setting with image annotations into machine translation. Their approach obviates the need to have bilingual sentences aligned with image annotations, a very restricted resource. An attention layer allows the transformer to incorporate a topic-image lookup table. Their approach achieves significant improvements over strong baselines. The reviewers and the authors engaged in substantive discussions. This is a strong paper which should be included in ICLR.